copLAB gene prevalence and diversity among Trinidadian Xanthomonas spp. black-rot lesion isolates with variable copper resistance profiles

http://orcid.org/0000-0002-0937-0046 Ramnarine Stephen DB Jr
Jayaraman Jayaraj
Ramsubhag Adesh adesh.ramsubhag@sta.uwi.edu
Life Sciences, The University of The West Indies, St. Augustine , St. Augustine , Trinidad
Ahmad Faheem
Electronic publication date: 2023 Jun 27
Publication date: 2023
Volume: 11
Electronic Location ID: e15657
Received 2023 Jan 11; Accepted 2023 Jun 7
Copyright: © 2023 Ramnarine et al.
Copyright year: 2023
Copyright holder: Ramnarine et al.
License: This is an open access article distributed under the terms of the Creative Commons Attribution License, which permits unrestricted use, distribution, reproduction and adaptation in any medium and for any purpose provided that it is properly attributed. For attribution, the original author(s), title, publication source (PeerJ) and either DOI or URL of the article must be cited.
License URL: https://creativecommons.org/licenses/by/4.0/

Keywords: Xanthomonas campestris pv. campestris, Xanthomonas sp., Copper resistance, copLAB, cop lab gene diversity

Funding: The University of the West Indies, St. Augustine, Trinidad Campus Research and Publication Fund CRP.5.APR17.45 This work was supported by the Campus Research and Publication Fund (CRP.5.APR17.45) of The University of the West Indies, St. Augustine, Trinidad awarded to Stephen DB Jr Ramnarine. The funders had no role in study design, data collection and analysis, decision to publish, or preparation of the manuscript.

==============================
Background

There has been limited exploration of copLAB genotypes and associated copper resistance phenotypes in Xanthomonas spp. in the southern Caribbean region. An earlier study highlighted a variant copLAB gene cluster found in one Trinidadian Xanthomonas campestris pv. campestris (Xcc) strain (BrA1), with <90% similarity to previously reported Xanthomonas copLAB genes. With only one report describing this copper resistance genotype, the current study investigated the distribution of the BrA1 variant copLAB gene cluster and previously reported forms of copper resistance genes in local Xanthomonas spp.

Methods

Xanthomonas spp. were isolated from black-rot infected lesions on leaf tissue from crucifer crops at intensively farmed sites with high agrochemical usage in Trinidad. The identity of morphologically identified isolates were confirmed using a paired primer PCR based screen and 16s rRNA partial gene sequencing. MGY agar amended with CuSO4.5H2O up to 2.4 mM was used to establish MIC’s for confirmed isolates and group strains as sensitive, tolerant, or resistant to copper. Separate primer pairs targeting the BrA1 variant copLAB genes and those predicted to target multiple homologs found in Xanthomonas and Stenotrophomonas spp. were used to screen copper resistant isolates. Select amplicons were sanger sequenced and evolutionary relationships inferred from global reference sequences using a ML approach.

Results

Only four copper sensitive/tolerant Xanthomonas sp. strains were isolated, with 35 others classed as copper-resistant from a total population of 45 isolates. PCR detection of copLAB genes revealed two PCR negative copper-resistant resistant strains. Variant copLAB genes were only found in Xcc from the original source location of the BrA1 strain, Aranguez. Other copper-resistant strains contained other copLAB homologs that clustered into three distinct clades. These groups were more similar to genes from X. perforans plasmids and Stenotrophomonas spp. chromosomal homologs than reference Xcc sequences. This study highlights the localisation of the BrA1 variant copLAB genes to one agricultural community and the presence of three distinct copLAB gene groupings in Xcc and related Xanthomonas spp. with defined CuSO4.5H2O MIC. Further characterisation of these gene groups and copper resistance gene exchange dynamics on and within leaf tissue between Xcc and other Xanthomonas species are needed as similar gene clusters showed variable copper sensitivity profiles. This work will serve as a baseline for copper resistance gene characterisation in Trinidad and the wider Caribbean region and can be used to boost already lacking resistant phytopathogen management in the region.

Introduction

Copper salts have been commonly used in chemical-based management schemes for bacterial and fungal diseases in agriculture for more than a century (Ayres, 2004; Lamichhane et al., 2018). Their consistent use was quickly followed by reports of reducing disease management efficacies from as early as the 1980s. This was shortly followed by the discovery of copper-resistant phytopathogens (Adaskaveg & Hine, 1985; Cooksey, 1987; Sundin & Bender, 1993; Goto et al., 1994)⁠. Among these were Xanthomonas spp. (Xsp) that form a major group of bacterial phytopathogens that affect multiple vegetable crops worldwide and are now known to have a high occurrence of copper resistance (Lamichhane et al., 2018)⁠. Xanthomonas campestris pv. campestris (Xcc), the causal agent of black-rot in crucifers (Vicente & Holub, 2013), is of particular concern due to the lack of up-to-date studies concerning this species in the Caribbean region. This pathogen causes significant yield loss in cruciferous vegetables in Trinidad and the southern Caribbean. Disease management in this country still depends heavily on chemical usage with little or no rotation of diverse chemicals. Continued application of chemical formulations with copper salts has been met with reduced disease management outcomes in Trinidad. Furthermore, the mono-cropping of solanaceous and cruciferous crops has exacerbated chemical exposure and pathogen enrichment in these farming districts (Ali, Ramsubhag & Jayaraman, 2021). These agricultural practices are not unique to Trinidad and have contributed to the dire consequence of the global occurrence of copper-resistant Xanthomonas spp. across six continents (Lamichhane et al., 2018).

Copper resistance phenotypes in Xanthomonas spp. are attributed to the presence of a plasmid-borne cop operon and survival at >0.8 mM CuSO4.5H2O (Marin et al., 2019). These plasmid-borne operons and homologs have been well described in numerous publications since the 1990s (Lim & Cooksey, 1993; Voloudakis, Bender & Cooksey, 1993; Behlau et al., 2011). Several studies have shown the presence of homologs of these cop genes and their extensive horizontal transfer among bacterial phytopathogens (Behlau et al., 2012, 2013)⁠. Copper-resistant Xanthomonas spp. and closely related Stenotrophomonas spp., are known to share ~90% nucleotide similarity among the copLAB genes implying horizontal transfer within these genera (Teixeira et al., 2008; Behlau et al., 2012). While cop operon gene content is not consistent in all plasmids, the copLAB genes are common and account for the major copper resistance phenotype (Behlau et al., 2011). The copL gene functions as a transcriptional activator of downstream cop gene expression in the presence of elevated intracellular copper ions (Pontel & Soncini, 2009; De Freitas et al., 2019)⁠. The copA gene encodes a multicopper oxidase and copB is predicted to bind Cu ions but may be involved in removal from the periplasm (Teixeira et al., 2008; Behlau et al., 2013). While the relationship between diverse copLAB genes and copper resistance levels is yet to be fully explained, understanding the changes in copLAB gene diversity within Xanthomonas spp. and related species in relation to disease management practices are of paramount importance⁠.

In Trinidad, copper-resistant Xcc was first reported by Lugo et al. (2013), who noted a positive correlation between resistant isolates and monoculture of cruciferous vegetables at agricultural sites. One of the copper-resistant Xcc isolates (BrA1) from the farming district of Aranguez contained a plasmid-borne copLAB gene cluster with very low nucleotide homology to published Xcc copLAB genes (Behlau et al., 2017)⁠. The report by Behlau et al. (2017) also highlighted the inability of current PCR amplification and nucleic acid hybridization-based diagnostic strategies to detect this “variant” form of copper resistance. The prevalence of the BrA1 variant copLAB gene cluster in Xcc populations on the island has not been established, nor are there detailed reports on the diversity of copper resistance genes among strains of this pathogen as a whole. The genomes of some local copper resistance Xcc and other Xsp were reported in Ramnarine, Jayaraman & Ramsubhag (2022), which revealed copper responsive elements further characterised in ongoing works (SDB. Ramnarine Jr, J. Jayaraman, A. Ramsubhag, 2023, unpublished data). The current study aimed at determining the presence of the BrA1 variant copLAB genes identified from local draft genomes and, attempt to capture other forms using primers designed against Xsp and Stenotrophomonas spp. homologs. Xcc and other Xsp were obtained from black-rot infected leaf tissue sampled at crucifer farms in four farming districts in Trinidad, inclusive of Aranguez where Xcc BrA1 was first isolated. Crop cultivation at these locations has been almost consistent for >30 years. PCR-based detection and sequence analysis coupled with evolutionary analysis was employed to determine the copLAB genetic diversity in these bacterial isolates. The copLAB genes characterized by Behlau et al. (2017) are distinguished in this study with the label “Variant” and are compared to previously reported genes contained in the NCBI database from Stenotrophomonas spp. and Xanthomonas spp.

Methods

Agricultural sites involved in the study with varying intensities of cultivation

Leaf samples of cabbage and cauliflower showing symptoms of black-rot infection were collected from two fields at each location indicated in Fig. 1. The farming districts of Maloney and Aranguez have >30-year history of agricultural land use, whereas Navet and, Bon Air have a shorter history (<15 years). Each agricultural district has been associated with crucifer cultivation, with a greater incidence of monoculture in the Navet district.

Figure 1 Agricultural districts sampled in Trinidad, Trinidad and Tobago, W.I.

Made with Vemaps.com. © 2023 Vemaps.

Sample collection and bacterial isolation

Leaf samples were stored at 4 °C, and bacteria were isolated within 24 h according to Lugo et al. (2013). Isolates similar to or with matching morphologies to Xanthomonas (yellow, convex shaped colonies) were selected after 48 h incubation on nutrient agar (NA). These isolates were labelled as the lesion-associated population. Further investigation into the distribution of copper-resistant bacteria in the Aranguez district was carried out in 2017 to determine the distribution of variant BrA1 copLAB genes between bacteria at an active and abandoned crucifer field. Phylloplane-associated and soil-borne bacteria were isolated from plant material and soil respectively from an active cauliflower field and abandoned plot approximately 300 m away. Isolates were obtained by plating dilution washes on NA. Morphologically unique colonies were selected after 48 h and labelled as the environmental reference population. In total 45 lesion-associated, and 199 environmental population isolates were retained and stored at −80 °C in 25% glycerol/Nutrient Broth.

Taxonomic assignment of bacterial isolates in the lesion-associated and environmental reference population

Lesion-associated isolates displaying similar morphologies to Xanthomonas were identified as XCC using primers (XCF-CGATTCGGCCATGAATGACT, XCR-CTGTTGATGGTGGTCTGCAA) targeting a highly conserved region of the Xcc hrpF gene (Park et al., 2004) or as a member of the Xanthomonas genus (Xsp) using primers targeting the hrpB6 gene (RST2-AGGCCCTGGAAGGTGCCCTGGA, RST3-ATCGCACTGCGTACCGCGCGCGA) (Leite et al., 1994), or using RT 16 F/R primers targeting the gumL gene (Pandey et al., 2016). Isolates with expected amplicons for XCF/R and RT 16 F/R primers were labelled as Xcc while those with RST2/3 and or RT 16 F/R bands were labelled as Xsp. Strains without amplification with any primer were not used in further analysis. Classification using this scheme was corroborated using partial 16s rRNA sequencing (Macrogen, Korea), which was also applied to environmental isolates. Sequences with QV >16 were selected and aligned using BLAST (BLASTN, https://blast.ncbi.nlm.nih.gov/) against the NCBI 16s rRNA database. Only copper-resistant environmental isolates were identified using 16s rRNA sequencing. While species-level identification was obtained for most isolates (Coverage >90% and %ID >98%), only identified genera were considered for consistency across the environmental dataset. WGS characterised local Xcc and Xanthomonas melonis (Xmel) isolates (Ramnarine, Jayaraman & Ramsubhag, 2022) were included as controls. Partial 16s rRNA Sanger-generated sequences can be accessed at https://doi.org/10.5281/zenodo.6795859.

Copper sensitivity profiling

Copper sensitivity screening was carried out according to Ramnarine, Jayaraman & Ramsubhag (2022) using CuSO4.5H2O concentrations ranging from 0–2.4 mM amended into Mannitol Glutamate Yeast Agar. Briefly, the copper sensitivity of isolates was categorized based on the MIC of CuSO4.5H2O of isolates after 48 h incubation on amended media at 28 °C. Copper-sensitive isolates only grew in <0.6 mM CuSO4.5H2O, copper tolerant strains survived concentrations between 0.6–0.8 mM while copper resistant isolates survived >0.8 mM. Amended media was prepared using a filter sterilized CuSO4.5H2O stock solution both prepared fresh on the day of use. A total of 48 h cultures of bacterial isolates were suspended in sterile water and 5 µL spot plate onto solid amended agar in triplicate.

copLAB gene detection in lesion-associated and environmental isolates

Primers were designed for the BrA1 variant copLAB gene sequences (Behlau et al., 2017) using reference homologs (similarity >= 60%) from the Xanthomonas and Stenotrophomonas genera available from GenBank. These were aligned using ClustalW in BioEdit v7.0.5.3 (Hall, 1999)⁠. Variant-specific primers targeting unaligned regions unique to the BrA1 variants were manually designed and screened for specificity using PrimerBlast (https://www.ncbi.nlm.nih.gov/tools/primer-blast/).⁠ Those predicted to amplify variant genes unambiguously were further screened using gDNA of Xcc strain BrA1, and the amplicons were Sanger sequenced (Macrogen, Seoul, Korea) before cross-referencing to the draft BrA1 genome contig containing the variant genes (GCA_002806765.1). Primers targeting aligned regions from reference homologs (similarity >= 60%) from the Xanthomonas and Stenotrophomonas genera available from GenBank were designed to target previously reported sequences. Primer pairs were screened for specificity using PrimerBlast (https://www.ncbi.nlm.nih.gov/tools/primer-blast/).⁠ These primers were in-silico verified to amplify those from plasmids, characterised genes from copper resistant Xsp and those from Stenotrophomonas chromosomes with identical synteny to copLABMGF copper resistant operons. All primers were synthesized at IDT, sequences and amplicon length are listed in Table 1.

Table 1 Primers designed for PCR amplification of previously reported copLAB genes.

Primers	Primer Sequence (5′–3′)	Amplicon size (bp)	Target	
SMXCOPLF	GGTGGTCGGATCAGATGAGG	405	Xanthomonad copL	
SMXCOPLR	GCCCGTGTCAGCCTC	
XEGVCOPAF	GTGCAAATGGATGGGATG	514	Xanthomonad copA	
XEGVCOPAR	GGATCATGGGTCGATGGAT	
XCCCOPBGF	GACCACAGTCAGATGGGACAC	744	Xanthomonad copB	
XCCCOPBGR	GGTACGAAGGCCGATGACCA	
copLSRF2	TGGCATCCCATCCTCTTTCG	311	BrA1 copL	
copLSRR2	ACAGGAATAGACGCACAGGC	
copASRF9	TGTCCTTAGTGGGACCGAGT	306	BrA1 copA	
copASRR9	CCTTGCTGAACCGTAAAGCG	
copBSRF2	TACCGACCTCAACCGTCTCT	533	BrA1 copB	
copBSRR2	GAACCAAGTGCGTAGACCGA	

Total genomic DNA was isolated from all lesion-associated and copper-resistant environmental isolates according to Wilson (2001) and quality was assessed using standard gel electrophoresis methods (1% agarose). PCR amplification using designed primers (Table 1) and molecular reagents from Bioland Scientific were carried out in 25 µL reactions as outlined in Ramnarine, Jayaraman & Ramsubhag (2022).

copLAB gene sequencing and evolutionary analysis of variants

Amplicons generated from primers targeting both variant and previously reported genes were gel purified using the Wizard® SV Gel and PCR Clean-Up System (Promega, Madison, WI, USA) and, sequenced on an ABI3730XL platform (Macrogen, Seoul, Korea). An overall QV score threshold of <16 was set for sequence rejection and end trimming. Pairwise blast using the BLASTN and TBLASTN (https://blast.ncbi.nlm.nih.gov/) algorithms were used to verify the homology with annotated copLAB reference sequences against the entire NCBI GenBank database. Final curated sequences, GenBank references, and full gene sequences from the Xcc BrA1 and other local Xsp genomes⁠ (Table S1) were submitted to ngphylogeny (https://ngphylogeny.fr/) for evolutionary analysis. Using this server, multiple alignment and curation were carried out using MAFFT (Katoh & Standley, 2013) and BMGE (Criscuolo & Gribaldo, 2010) respectively. Alignments were then submitted to FastTree (Price, Dehal & Arkin, 2010) for maximum likelihood phylogenetic reconstruction with 1,000 bootstrap branch support (Felsenstein, 1985). The bootstrap consensus tree for each dataset was visualized and edited in TreeGraph 2 (Stöver & Müller, 2010)⁠. Alignments of copLAB sequences were also analysed in MEGA 11 (https://www.megasoftware.net/) to identify variable sites. Sanger-generated sequences for amplified copLAB genes can be accessed at https://doi.org/10.5281/zenodo.6795859.

Statistical inference

Statistical analysis was carried out on lesion associated and environmental isolate datasets using RStudio (2023.03.0+386; R Studio Team, 2023) and R version 4.2.3 (R Core Team, 2023). Statistical tests included Anova, TukeyHSD post-hoc test and Pearson’s Chi-squared Test from the R stats package (v4.2.3).

Results

copLAB gene prevalence in copper-resistant lesion-associated Xanthomonas spp

The 45 lesion-associated isolates were distributed by agricultural district as follows: Aranguez = 12, Maloney = 5, Navet = 24, Bon Air = 4. These are represented in Table 2 which includes species-level identification, copper sensitivity profiles and max CuSO4.5H2O MIC. Most isolates from this population were copper resistant (78%) but only 31% were Xcc. There were <7% tolerant and only one sensitive isolate obtained but profiles of five isolates could not be determined. Most (80%) of the copper-resistant isolates grew in the presence of 2.4 mM CuSO4.5H2O with the lowest MIC for this category being 1.2 mM. Among the Xanthomonas species there were significant differences in CuSO4.5H2O MIC means (P = 0.0176) but only between Xsp and Xcc (P = 0.0134). However, no significant difference was observed between MIC’s, cop genotype and sample location.

Table 2 Local Xanthomonas copper sensitivity profiles, PCR-based identification, and cop genotype.

Isolate	Location	Species	cop genotype	CuSO4.5H2O MIC (ppm)	CuSO4.5H2O MIC (mM)	Cu Sensitivity	
AR1PC2	Aranguez	Xcc	Undetected	200	0.8	T	
Cf4B	Variant	-	-	-	
Ar1BCA2	400	1.6	R	
BrA1	300	1.2	
Ca3B	600	2.4	
Cf1A	600	2.4	
Cf3A1	600	2.4	
Cf3C	600	2.4	
Cf4B1	600	2.4	
Cf5A2	600	2.4	
Cf5B	600	2.4	
Cf6A1	600	2.4	
FRL2D3	Bon Air	Xcc	Traditional	-	-	-	
FRL2G4	-	-	
FRL1C1	Xsp	400	1.6	R	
FRL2G5	600	2.4	
DMCE	Maloney	Xmel	Traditional	-	-	-	
DMCA	600	2.4	R	
DMCX	600	2.4	
DMCK*	Undetected	600	2.4	
CCB4	Xsp	Traditional	-	-	-	
CaNP6B	Navet	Xcc	Traditional	500	2	R	
CNP1E	300	1.2	
CNP2A*	Undetected	400	1.6	
CNP3C*	400	1.6	
CaNP1C	100	0.4	S	
CNP4A	200	0.8	T	
CaNP6A	Xmel	Traditional	600	2.4	R	
CaNP5B	600	2.4	
CaNP1D	Undetected	200	0.8	T	
CaNP3B	Xsp	Traditional	600	2.4	R	
PNP25	600	2.4	
PNP26	600	2.4	
PNP34	600	2.4	
PNP39	600	2.4	
PNP44	600	2.4	
PNP49	600	2.4	
PNP54	600	2.4	
PNP58	600	2.4	
PNP62	600	2.4	
PNP63	600	2.4	
PNP64	600	2.4	
PNP72	600	2.4	
PNS3*	Undetected	600	2.4	
Note:

* Copper-resistant isolate with no detected copLAB genes for primers used.

Xmel, Xanthomonas melonis. cop genotype—Traditional, similarity to previously reported copLAB sequences; Undetected, PCR negative result; Variant, BrA1 variant copLAB genes. CuSO4.5H2O MIC—represented as ppm and mM, max concentration as determined using media screens. Cu Sensitivity—R, Resistant; T, Tolerant; S, Sensitive.

Figure 2 only represents copper resistant isolates (35) which survived four CuSO4.5H2O concentrations represented in Table 2, 1.2 mM (2), 1.6 mM (4), 2 mM (1) and 2.4 mM (28). Only isolates from Aranguez (11) contained all three variant BrA1 copLAB genes (Variant, Fig. 2). Previously reported copLAB genes (Traditional, Fig. 2) were found in Xsp, Xmel and Xcc strains from Maloney, Bon Air and Navet. The BrA1 Variant copLAB primers were able to amplify individual copLAB genes in WGS characterised local Xcc. Primers targeting Xanthomonad copLAB genes amplified those in characterised Xmel strains described in Ramnarine, Jayaraman & Ramsubhag (2022). No significant difference in MIC’s were observed between variant and previously reported cop genotypes. However, there was a significant difference in cop genotypes (P = <2e−16) by location, specifically when comparing Aranguez to the other three districts.

Figure 2 copLAB gene prevalence in local copper resistant Xanthomonas.

Traditional refers to previously reported copLAB genes. Only copper-resistant isolates (35) are represented in this figure. WGS characterised Xcc and Xmel strains are also represented here. Copper resistant sample sizes: Aranguez = 10, Maloney = 3, Navet = 20, Bon Air = 2.

Two Xcc and one Xsp from Navet and, two Xsp from Maloney were characterised as copper resistant, but PCR amplification was not able to detect any copLAB gene. Only one sensitive (Xcc CaNP1C) and three tolerant (Xcc Ar1PC2 and CNP4A and Xmel CaNP1D) strains were isolated and were all copLAB PCR negative. All 45 isolates and associated isolation location and PCR prevalence data are given in Table S2. In brief, most of the 45 Xanthomonas isolates were copper resistant (80%) up to 2.4 mM CuSO4.5H2O. From these isolates, only Xcc strains from the Aranguez district contained the BrA1 variant copLAB gene cluster. Previously reported copLAB genes were also identified in copper resistant isolates from other agricultural districts using the primers designed in this study. However, four copper resistant strains, including Xcc, were PCR negative for copLAB genes. Significant differences in CuSO4.5H2O MIC were observed between Xcc and Xsp strains and with the BrA1 variant cop genotype in Aranguez compared to the other districts.

BrA1 variant copLAB gene prevalence among copper resistant environmental phylloplane and soil associated bacteria from the Aranguez district

Differences in the proportions of isolates characterised as copper sensitive, tolerant and resistant in the environmental population are summarised in Fig. 3. Full isolate profiles are given in Table S3. Greater proportions of resistant isolates were obtained from the phylloplane and soil of the cultivated field, with the highest proportions from the latter environment (57%) (Fig. 3B). While copper-resistant isolates were found in all environments and higher proportions were observed among soil isolates (Fig. 3B). Copper tolerant and sensitive isolate proportions were higher from the phylloplane and soil of the abandoned plot respectively (20% vs. 48% and 20% vs. 49%). Overall, of the 199 isolates screened, 78 were copper sensitive, 46 tolerant and 75 were resistant. Pearson’s Chi-squared test showed a significant association between copper sensitivity and, the sampling environment and location (P = 0.0006). Using Pearson residuals, a positive correlation was observed between copper tolerant bacterial strains and the phylloplane environment of the abandoned field. This was also seen with copper resistant strains and the soil environment of the cultivated field. In summary, there were statistically significant differences in copper sensitive isolates from the phylloplane and soil of a cultivated and abandoned field. From this there were greater positive correlations with copper resistant isolates and the cultivated field.

Figure 3 Proportional differences in copper sensitivity of phylloplane (A) and soil (B) environmental bacterial isolates from a cultivated and an abandoned field in the Aranguez agricultural district.

Proportions are given as percentages of total isolates per field per environmental sample. The number of isolates is given in parentheses above each pie chart.

The 75 copper-resistant isolates from the environmental population were screened for the variant copLAB genes and identified using 16s rRNA gene sequencing (Table 3). Interestingly, no copper-resistant Xsp were identified within this sub-group of environmental isolates. Notable is the higher diversity in soil and phylloplane from the cultivated field and the presence of Stenotrophomonas spp. isolates (Table 3) in all locations except the phylloplane of the abandoned field. More isolates from this genus were obtained from the cultivated field which also had a greater number of Pseudomonas spp., Achromobacter spp. and unidentified isolates. However, no copper-resistant isolate in the environmental population contained the BrA1 variant copLAB gene.

Table 3 Copper-resistant bacterial isolate counts from an environmental population originating in the Aranguez agricultural district.

	Phylloplane	Soil	
Genus	Abandoned plot	Cultivated field	Abandoned plot	Cultivated field	
Achromobacter	0	0	0	3	
Acinetobacter	0	4	0	3	
Bacillus	3	5	0	0	
Enterobacter	0	0	7	0	
Klebsiella	0	0	3	0	
Pseudomonas	0	9	4	4	
Serratia	0	1	0	3	
Shigella	0	0	0	1	
Sphingomonas	1	0	0	0	
Stenotrophomonas	0	3	2	6	
Note:

Bacterial counts for each identified genus from a sub-group of the environmental isolate population is shown is this table.

Evolutionary analysis of copLAB genotypes

Sequences from local X. sp. isolates were similar (>96%) to previously reported cop sequences from Xanthomonas plasmids or Stenotrophomonas spp. chromosomes. Furthermore, these sequences were not clustered with coh chromosomal homologs further establishing confidence in their designation as cop sequences. In line with established similarity patterns in global strains, copA (Fig. 4B) and copB (Fig. 4C) genes from local isolates appear to be more closely related than copL (Fig. 4A). In each phylogeny, three distinct groupings of PCR amplified copLAB genes from local isolates were observed. Group 1 and 3 copL sequences, were most distantly placed when compared to all other sequences. Both groups consisted of genes ~93.5% similar to each other. Isolates in these groups were all from Navet. Group 1 strains had a CuSO4.5H2O MIC of 400 ppm with Group 3 being 600 ppm. Notably, these identical Group 1 sequences were obtained from two Xanthomonas species (Xcc and Xsp). When compared to X. euvesicatoria pLMG930.1, four variable sites in Group 1 copL sequences from both Xcc and Xsp were identified. Group 3 sequences were identical for the aligned region but did not cluster with a previously reported sequence. However, when compared to Group 1, there were 22 variable sites in the aligned sequences in Group 3. Interestingly, in the BrA1 variant Group 2, four variable sites were detected when compared to the BrA1 draft genome copL sequence. However, these variations were only present in Xcc Ca3B and Cf6A1. Overall, the copL sequences in all three groups contained 147 variable sites.

Figure 4 Phylogenetic reconstruction of amplified partial copL (A), copA (B) and copB (C) sequences from local copper-resistant Xanthomonas isolates.

Sequences from PCR amplifications are highlighted in blue, those from local isolate draft genomes are given in purple, and other sequences originate from GenBank reference genomes of Xanthomonas spp., Stenotrophomonas spp. and Xanthomonas plasmids. Chromosomal copLAB homologs (coh) are highlighted in orange boxes while the BrA1 variant cop genes are in blue boxes.

The copA sequences formed two distinct groups, with Group 1 consisting of the BrA1 variant genes. While Group 2 comprised Xsp and Xcc isolates from Navet, isolates in both groups had a CuSO4.5H2O MIC of 600 ppm. The Xcc CaNP6B sequence in this group almost entirely (99%) aligned with 99.8% similarity to multiple copA homologs in Stenotrophomonas spp. and Xanthomonas spp. plasmids. The Xsp sequences in Group 2 were identical to a partial copA CDS from X. perforans (MK616431) while local Xcc variant sequences clustered with the BrA1 variant and a Stenotrophomonas sp. copA gene. These variant sequences in Group 1 did not show the same trend as copL (Fig. 4A Group 2) as all sequences were identical. Furthermore, when compared to other sequences in Group 2, Xcc CaNP6B shared 15 variable sites with the Xsp sequences in this group (which were all identical). Overall, both groups shared 153 variable sites.

Previously reported copB sequences clustered into two closely related groups, 2 and 3 (Fig. 4C). Group 2 contained a single sequence from Xcc CaNP6B which was similar to sequences from the local Xmel CaNP6A genome but identical to multiple Stenotrophomonas spp. homologs (Similar to Fig. 4B, Group 2). Notably, the copL and copA genes from this Xcc strain did not show this clustering pattern. Group 3 sequences clustered with the well-referenced copL sequence from X. euvesicatoria citrumelonis (HM579937) showing 99.5% similarity at 99% coverage. Isolates from all groups had a CuSO4.5H2O MIC of 600 ppm. In Group 1, the BrA1 variant sequences were all identical (as seen in Fig. 4B Group 1). Despite their separate group designation, the closely placed Xcc sequence and Xsp sequences in Groups 2 and 3 (Fig. 4C) were compared. Xcc CaNP6B copB shared 46 variable sites with Group 3 sequences. Within Group 3, however, two groupings based on four variable sites were observed. These consisted of PNP25 and PNP34 and, PNP54 and PNP62. Overall, the sequences from all three groups shared 210 variable sites. As noted with the previous Xcc BrA1 study, variant sequences from local strains clustered separately from others. These variant sequences also clustered with and were identical to cop homologs from Stenotrophomonas sp. WZN-1 (CP021768.1). Variant copLAB sequences are represented in Fig. 4A (Group 2) and Figs. 4B and 4C (Group 1), respectively in blue boxes.

Discussion

This study demonstrated that 70% of Xcc strains and 78% of the overall Xanthomonas population had copper resistance up to a high MIC of 600 ppm. Drastically, only one Xcc strain was copper sensitive. Furthermore, three cop genotypes were identified and associated with different copper resistant MIC. One previous study established copper sensitivity profiles of Xcc strains isolated between 1999 and 2003 (Lugo et al., 2013). Copper sensitive and resistant (59% of isolates) strains were obtained in that study. This trend indicates that copper resistance and tolerance levels in Xcc have increased since 2003, in line with expectations given that no drastic change in copper-based chemical usage guidelines have taken place since. In Trinidad, the agricultural use of chemicals is not strictly regulated beyond the point of sale. Application rates and concentration depend heavily on the discretion of farmers at the field level. In one case, the indiscriminate and monotonous use of copper-based fungicides was previously reported at most of the farming sites in Trinidad (Ali, Ramsubhag & Jayaraman, 2021). Lugo et al. (2013)⁠ noted the correlation of higher proportions of copper resistant Xcc with long-term use of copper sprays in Trinidad. With some overlap in sample locations, this study further evaluated Xanthomonas isolates from Aranguez, Maloney, Navet and Bon Air to determine copLAB genotypes. Copper resistance has been linked to plasmid-borne copLAB genes in Xanthomonas spp. in numerous studies (Pontel & Soncini, 2009; Behlau et al., 2013; Bondarczuk & Piotrowska-Seget, 2013). Thus, all copper resistant Xanthomonas isolates in this study were expected to contain plasmid-borne copLAB homologs.

Sequencing partial copLAB genes of the BrA1 variant and previously reported forms in local Xanthomonas strains illustrated separate groupings in the deduced phylogenies. The BrA1 variant genes from local Xcc strains clustered separately from other previously reported sequences except for one Stenotrophomonas chromosomal homolog (CP021768.1). Apart from the latter observation in this study, this clustering pattern was noted by Behlau et al. (2017). Overall, the closest related reference clades to local sequences originated from Xanthomonas spp. plasmids and Stenotrophomonas spp. chromosomes, but notably, did not include Xcc. These Xcc reference sequences were most distantly placed in all phylogenies. This possibly indicates an origin of copLAB genotypes from other Xsp and Stenotrophomonas sp. within the local Xcc ecosphere. Horizontal transfer and the homology of copLAB genes between Xanthomonas spp. and Stenotrophomonas spp. have been documented in other studies (Behlau et al., 2013; Lai et al., 2021). Notably, diverse copLAB genotypes within Xcc species are not well-reported and often rely on homology-based searches in reference to established forms from other species.

Interestingly, all copper resistant Xcc from Aranguez contained the BrA1 variant. This genotype appears to be strongly associated only with this species and only at this location. Of note is the 100% identity to a global Stenotrophomonas sp. chromosomal operon but the absence of this variant in any copper resistant Stenotrophomonas strains from environmental isolations. Also remarkable is the apparent lack of drift into other agricultural districts. However, this will need to be reassessed with a newer and more extensive sampling population. The gene neighbourhood of the BrA1 variant copLAB operon and the identical reference both contain mobile elements (SDB. Ramnarine Jr, J. Jayaraman, A. Ramsubhag, 2023, unpublished data), and its localization on a plasmid further adds evidence of interaction with another bacterial species and mobile elements. That is, the movement of this variant into Xcc appears to be indirect.

Two major groups of copA and B sequences were observed in local Xanthomonas and Xcc strains. With copL sequences, three groups were observed. Furthermore, there are also at least two unique copLAB clusters within local copper resistant Xcc. Two strains from Navet did not produce amplicons and thus potentially contain a novel variant. Another strain from Navet contained diverse copL and A genes with partial operon homology to more than one reference sequence. The latter case implies some measure of recombination which was not fully assessed due to the length of sequences obtained. Studies have noted that environmental exposure to copper chemicals can impact bacterial diversity and the persistence of copper resistance genes in strains (Araújo et al., 2012). One study noted that the continuous selection pressure of copper ions was linked to the presence of cop genes in environmental Xanthomonas sp. (Roach et al., 2020)⁠. Copper resistance is a complex trait in different bacterial strains and the persistence of resistance genes can be attributed to horizontal transfer of plasmid-borne copper resistance elements (Altimira et al., 2012) or enrichment of existing stress response factors within strains (Teelucksingh, Thompson & Cox, 2020).

While it is largely unknown what advantages these variant genes may have in the local Xanthomonas population, it can be presumed that the persistence of certain copLAB genotypes provides advantages over others within this species. This was observed as most genotype groups survived CuSO4.5H2O MIC up to 600 ppm except for one group of Xcc and Xsp which survived up to 400 ppm. The impact of cop genotype on these MIC values can also be investigated via variable sites in the copA and B genes. The copL gene is not predicted to be protein coding and its exact function in regulation is largely unknown but copA and B are copper binding proteins. The amplified sequences used in this study covered copper binding motifs from both genes (SDB. Ramnarine Jr, J. Jayaraman, A. Ramsubhag, 2023, unpublished data). It is thus assumed that these variable sites may play a role in MIC differences due to changes in copper binding capacity. One study evaluated mutations in the copA gene binding motifs and demonstrated that nucleotide changes affected overall Xcc strain MIC against copper (Hsiao et al., 2011). In the wider environment, consistent application of copper appears to enrich for copper tolerant and resistant bacteria in the soil. Interestingly, the persistence of resistant and tolerant environmental bacteria was even noted in soil and plant material from an abandoned agricultural field. Not only are there cop genotypes in Xanthomonas strains allowing for survival up to 600 ppm CuSO4.5H2O, after agricultural activity has stopped resistant strains persist in the environment. While older studies noted that fields without a consistent use of copper-based chemicals contained copper susceptible Xanthomonas (Adaskaveg & Hine, 1985; Lugo et al., 2013), this relationship appears to not revert once agricultural activity has ceased. Thus, for environments like those sampled in districts like Aranguez, the resistance phenotype is maintained in abandoned fields. Unfortunately, this adds to the complex nature of copper resistance in bacterial populations under the influence of human activity. The persistence of copper resistance further impacts disease and resistance management, proper agrochemical consumption in a largely unregulated space and, future agricultural land use.

Anecdotal data from local farmers indicates that black-rot in cabbage usually begins late in the cropping stage and the progression slows with copper sprays but eventually overwhelms the plant despite continuous applications. Understanding the different cop genotypes present in local Xanthomonas strains is an essential first step in resistance management as at least one group has been associated with a lower CuSO4.5H2O MIC in this study. Furthermore, the potential for novel cop genotypes exists at least in one agricultural district. The occurrence of novel variants within specific agricultural areas in Trinidad and the barriers which prevent their spread to other areas and related genera needs to be explored further. This is especially important in the Aranguez and Navet districts in the face of changing land use and far-reaching impacts of climate change.

Conclusion

This is the first study to report on the distribution of Xanthomonas spp. copLAB genes both in Xcc and Xsp. isolates from agrochemical contaminated sites in Trinidad. The variant copLAB genes first identified in a single local Xcc strain from Aranguez (Behlau et al., 2017) were seen to only occur in Xcc and only those from that same farming district. Two Xcc strains from another district did not yield amplicons for primers designed to target multiple cop homologs from Xsp plasmids and Stenotrophomonas sp chromosomes and another contained genes more similar to other local Xsp genotypes and worldwide plasmid sequences. Phylogenetic clustering for each individual copLAB gene indicated at least three potential gene clusters where one is unique in its gene complement homology. The closely related Xsp also serves as a benchmark for local copper resistant copLAB gene diversity necessary in future studies on the impact of disease management on resistance, shifts in Xanthomonas populations and changes in copLAB gene dynamics. Furthermore, all identified cop genotypes were associated with a max CuSO4.5H2O MIC. This updated evaluation on copper sensitivity of Xcc and Xsp from leaf lesions indicated an increase in copper resistant and tolerant strains in agricultural fields from 2003–2017. Lesion-associated strains were seen to survive CuSO4.5H2O concentrations up to 2.4 mM. The increase is not surprising given unchanged and non-standardized copper-based chemical usage in Trinidad. More concerning is the persistence of copper-resistant environmental bacteria observed in abandoned field soil, pointing to populations enriched for resistance after chemical applications have ceased. This study serves as an important record of changing copper sensitivities, not only among Xcc but also with lesion-associated Xsp. Furthermore, it is the first attempt at characterising copper resistance copLAB genotypes among these isolates and has demonstrated the tight association of a variant cluster to its original isolate’s location. Furthermore, despite 100% identity to a gene cluster in a global Stenotrophomonas sp strain, this cluster was not found in local isolates of this genus. Additionally, the potential for more diverse copLAB genotypes is possible in at least two Xcc strains. The clustering of local and worldwide copLAB sequences and, the lack of the variant cluster in the related Stenotrophomonas genus further highlights the complex dynamics behind the origin of Xcc copLAB genotypes.

Supplemental Information

Supplemental Information 1 Supplemental Tables.

Click here for additional data file.

We thank Mr Omar Ali (University of the West Indies, St. Augustine) for his critical manuscript review and advice on statistical methods.

Additional Information and Declarations

Competing Interests

Author Contributions

Field Study Permissions

Data Availability

The authors declare that they have no competing interests.

Stephen DB Jr Ramnarine conceived and designed the experiments, performed the experiments, analyzed the data, prepared figures and/or tables, authored or reviewed drafts of the article, funding Acquisition, and approved the final draft.

Jayaraj Jayaraman conceived and designed the experiments, prepared figures and/or tables, funding Acquisition, and approved the final draft.

Adesh Ramsubhag conceived and designed the experiments, analyzed the data, prepared figures and/or tables, and approved the final draft.

The following information was supplied relating to field study approvals (i.e., approving body and any reference numbers):

No approval was needed to collect diseased plant specimens from farmers. These farmers did not give consent for their names to be released in this publication

The following information was supplied regarding data availability:

The data is available in the Supplemental Files and Zenodo:

Stephen DBJR Ramnarine. (2022). 16s rRNA and copLAB partial sequences of copper resistant Xanthomonas spp. isolated from agrochemical impacted crucifer fields in Trinidad (Version V1) [Data set]. Zenodo. https://doi.org/10.5281/zenodo.6795859.

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
