# Peer review of "copLAB gene prevalence and diversity among Trinidadian Xanthomonas spp. black-rot lesion isolates with variable copper resistance profiles"

_PeerJ, doi:10.7717/peerj.15657_

## Round 0.1 · original submission · Major Revisions

We have received the reports from our reviewers on your manuscript. Based on the advice received, I feel that your manuscript needs to address the reviewer’s comments, which are given below.

Reviewer 1 ·

Basic reporting

Copper-containing agrochemicals are widely used across the world and increasing evidence are reported to alert high occurrence of copper resistance in Xanthomonas. Ramnarine et al. reported the prevalence and genetic diversity of copper resistance operon copLAB across Xanthomonas isolates obtained in Trinidad and, for the first time, demonstrated the potential copper resistance issue around the region. The data generated in this study is very valuable but unfortunately poorly described in this manuscript. I really would like to see the data/result in this work published but the overall manuscript needs major revision to re-organize the structure, language, and data illustration.


Abstract: the current abstract need to be revised to make the paragraph more logical. Please use “in this work” or “this study” instead of “current study” as the later one will make the point confusing. I would suggest follow the guideline here: (1) brief background and current report; (2) a clear and concise statement summarizing the finding in this work; (3) more detailed description about experiment design (strain isolation), approach (sequencing) and finding (prevalence and diversity); (4) significance and application.


Language: the overall language and terminology use of the manuscript needs to be thoroughly polished as since some sentences are hard to understand. Below are some examples:

Please use “previously reported copLAB” or “wild-type copLAB” instead of “traditional copLAB” in the manuscript.
Line-103, the current study -> this study
Line-135, please spell out the abbreviation for “NA” (Nutrient agar?) when used.
Line-187, please use “evolutionary analysis” instead of “Phylogenetic” as the analysis performed is mainly focused on gene variants and not linked to bacterial phylogeny.

Proof-reading by a native-level English speaker would improve the content throughout.

Experimental design

Methods: please make more sections for Methods as currently too much contents are merged in the same paragraph and the descriptions are mixed together. I would suggest making Methods into these sections: (1) Agricultural sites involved in this study (2) Sample collection and strain isolation (please specify the specific sample types and what strains were isolated) (3) Taxonomy or species identification by 16S sanger sequencing (4) Copper sensitivity profiling (5) copLAB genes detection and variants sequencing (6) Evolutionary analysis of copLAB variants.

Validity of the findings

Results: Result need to be revised to better organize and highlight the data and conclusion obtained in this work, some suggestions here: (1) please provide sub-title for each section of the result; (2) please make a brief summary at the end of each section, such as “In brief”, “Shortly”, “Together”; (3) Figures are not properly cited in some paragraphs, for example, Figure-5 need to be cited in Line-251 to 268.


Additional analysis: the evolutionary analysis in Figure-5 is very interesting and I would like to see more related analysis. The authors could further explore (1) analysis of the hot spot of mutation along genetic variants: it would be great if the authors could point out the frequency of specific mutations/variants in the copLAB operon with annotation of functional domain which will be helpful to figure out potential underlying mechanism.


Others:
Figure-4: Better to use a table to summarize the data here instead of bar-plot
Figure-5: Please put panel indicator A, B, C at the top left of the panels instead of top right

·

Basic reporting

The submitted manuscript describes the isolation os Xanthomonans strains from farming lands in Trinidad and the characterization of the copper-resistance profiles of these isolates, including both MIC measuments and genetic analysis of the copLAB copper-resistance genes, formely pointed in the literature as determinants of this phenotype.

Although the article is generally well-written, with only minor issues in English grammar left for correction, I find it hard to follow as most of the description refers to groups that are not marked properly in the phylogenies presented.

Still, the article covers the preceding literature in an adequate manner, with proper and up-to-date citations and is fairly objective in its intention to document observations of copper resistance and may be suitable for presentation after improvements in presentation and inclusion of some additional data.

Experimental design

The proposed design is appropriate for the study's goal, and this goal is in line with the requirements for originality and relevance, even if the manuscript is mostly targeted for a specific audience. Its greatest shortcome, though, is the presentation of results and the discussiion, both of which must be greately improved.

It is also important to consider better use of statistical toolls while interpreting the differences between sample sets. In this regard it is notable the the observed counts of resistant isolates were presented by not statistical test was made to compare the number of resistant isolates in different isolation locations. Also, possible correlations with the organismal phylogeny. which was not presented, should be explored because more closely related isolates could display similar resistance patterns even if the resistance determinants are carried by mobile elements and plasmids.

One methodological aspect that should be presented in the text is the procedure for assiging resistance phenotype for isolates: although a reference is given, this part of the procedure is so critical to this work that a short summary should be presented in this manuscript.

Validity of the findings

Overall the findings are interesting and seem corrrect but, as poiinted out above, rewriting and remodelling / improvement of figures and ables is required. For instance, photos of at least some of the gels with the bands from PCR amplification should be added to the supplment to allow evalution of the results.

Most importantly, all gene phylogenies should be presented in the same scale and oriented in a similar manner to allow for easy idnetification of congruent or equivalent clades. A phylogeny based on the 16S RNA gene sequences used for classification should also be provided, to allow comparison to the gene phylogenies.

All figures and legends must also be rreviewed to make sure they include all information required for their understanding, dispensing the need to search the main text for clarification. In this respect, a short explanation os each columns must be present in table legends.

Additional comments

I believe the manuscript presents valuable data but needs to undergo extended revision to achieve the format and clarity required by the journal standards.

---

## Round 0.2 · accepted · Accept

The authors have addressed both reviewers' comments within the revised version of the manuscript. Now the manuscript has been significantly improved, and I am glad to accept it.

Reviewer 1 ·

Basic reporting

Appreciation to all the authors for making the additional revision and improvements. I am very
happy to see my concerns/suggestions have been fully addressed.

I would recommend this manuscript for publication at PeerJ.

Experimental design

no comment.

Validity of the findings

no comment.

Additional comments

no comment.